# A New Global Bathymetry Model: STO_IEU2020

**Diao Fan** , **Shanshan Li \***, **Jinkai Feng, Yongqi Sun, Zhenbang Xu and Zhiyong Huang**

Information Engineering University, Zhengzhou 450001, China
* Correspondence: zzy_lily@sina.com

**Abstract:** To address the limitations in global seafloor topography model construction, a scheme is proposed that takes into account the efficiency of seafloor topography prediction, the applicability of inversion methods, the heterogeneity of seafloor environments, and the inversion advantages of sea surface gravity field element. Using the South China Sea as a study area, we analyzed and developed the methodology in modeling the seafloor topography, and then evaluated the feasibility and effectiveness of the modeling strategy. Based on the proposed modeling approach, the STO_IEU2020 global bathymetry model was constructed using various input data, including the SIO V29.1 gravity anomaly (GA) and vertical gravity gradient anomaly (VGG), as well as bathymetric data from multiple sources (single beam, multi-beam, seismic, Electronic Navigation Chart, and radar sensor). Five evaluation areas located in the Atlantic and Indian Oceans were used to assess the performance of the generated model. The results showed that 79%, 89%, 72%, 92% and 93% of the checkpoints were within the ±100 m range for the five evaluation areas, and with average relative accuracy better than 6%. The generated STO_IEU2020 model correlates well with the SIO V20.1 model, indicating that the proposed construction strategy for global seafloor topography is feasible.

**Keywords:** satellite altimetry; gravity anomaly (GA); vertical gravity gradient anomaly (VGG); seafloor topography (ST); modeling strategy

## 1. Introduction

The oceans account for about 71% of the earth's surface area [1], and are extremely rich in natural resources. On 1 January 2016, the United Nations proposed and launched the *2030 Agenda for Sustainable Development*, recognizing that a clean, healthy, transparent and predictable ocean is necessary for sustainable human development. However, people's understanding of the ocean remains largely insufficient. This vast frontier could offer untapped natural resources and major discoveries in future, which means that observing, measuring and understanding the ocean floor could further strengthen economic development.

The exploitation and utilization of the ocean and marine resources are highly dependent on the detection and cognition of the ocean. As the primary means of observing and understanding the oceans, seafloor topographic surveys play a fundamental role in the utilization of subaqueous resources, conservation of the marine ecological environment, advancements in marine science and technology and the protection of marine rights and interests. At present, seafloor topographic surveying mainly includes ship-based [2,3], submarine-based [2], satellite-based [4] and Airborne Lidar Bathymetry (ALB) [5]. Facing the whole ocean, ship-based hydrographic surveys have high accuracy, but are inefficient, and result in unevenly distributed surveys—especially in the southern hemisphere. ALB has high measurement efficiency, and is mainly used to measure shallow sea environments with good water quality [6]. However, it has limited topographic mapping capabilities in deep seas or sea areas, which limit beam propagation. Developing approaches and techniques that would accurately, efficiently and economically generate seafloor hydrographic data, particularly in remote and hard-to-reach sea areas, is incredibly important for managing and protecting marine resources and constructing the global seafloor topography (ST).

For more than 100 years, national organizations have been exploring ways to map the seafloor [7]. It was not until the 1990s when altimetry satellite data (e.g., GEOSAT and ERS-1), through geodetic missions, were accumulated and applied, and that global seafloor topographic mapping technology made significant progress and breakthroughs [8]. At present, satellite altimetry has become the primary method in charting the global ST. Many institutions have released a number of global ST models derived from satellite altimetry gravity data, such as the ETOPO1 model [9], SIO series model [10], GEBCO series model [11], SRTM+ series models [8,12] and BAT_WHU2020 model [13]. However, there are several shortcomings in global ST model construction:

The gravity element for ST construction is usually singular, which means that the input gravity field elements are limited. In general, the input gravity element used for almost all global ST construction is either gravity anomaly (GA) or vertical gravity gradient anomaly (VGG), except in a few cases where multivariate gravity data are applied to global ST modeling. Previous studies have confirmed that different gravity elements have their respective advantages in ST inversion depending on bathymetric depths and seafloor terrain [14,15]). Using Multivariate gravity data could improve the reliability of the ST construction, particularly in certain subaqueous landforms.

Some bathymetric data used in ST models are highly dependent on other ST models, causing weak model independence and performance. For example, most depth information in SRTM30+ are derived from SIO V11.1 ST; depth information in GEBCO_2019 and GEBCO_2020 are primarily from the SRTM15+ V1.0 model and the SRTM15+ V2.0 model, respectively. Most bathymetric deep-sea data in the ETOPO1 model are from SIO's early ST models.

At present, almost all global ST models only use one inversion method, and the ability to recover ST in certain complex situations can be weak. Studies have shown that the applicability and suitability of different ST inversion methods may vary significantly. For example, regression analysis is more suited for sea areas with more bathymetric data and uniform distribution than areas with sparse bathymetric data [16,17]. Frequency-domain inversion (i.e., admittance function) are often applied to ST in regular rectangular areas [18,19], but have poor results for the boundary areas of sea and land.

To address these limitations in global ST construction, we redesigned the global ST modeling strategy by considering the efficiency of ST construction, applicability of inversion methods and inversion advantages of different sea surface gravity elements. Using part of the South China Sea as the research area, we comprehensively analyzed the ST modeling strategy and evaluated its feasibility. Using SIO V29.1 GA and SIO V29.1 VGG, we recovered the global ST data (STO_IEU2020) and fused depth measurement results from multiple sources, including single beam and multi-beam bathymetric data and depths obtained via the seismic method, the Electronic Navigation Chart (ENC) and radar sensor. The ETOPO1 model, DTU18 model and SIO V20.1 model were introduced to evaluate the accuracy of the STO_IEU2020 model with preset external measured depth data as a reference.

## 2. Seafloor Topography Inversion Method and Construction Strategy

### 2.1. Seafloor Topography Inversion Method

The ST inversion method, based on sea surface gravity data, mainly includes the frequency-domain [18,20], regression analysis [10,21,22], Gravity-Geologic Method (GGM) [23–25], space domain [26,27] and simulated annealing (SA) [28,29]. Based on previous studies and our experiments, external bathymetric data is usually required to supplement non-inversion band ST information. This means that having a dense and uniform distribution of bathymetric data will significantly impact the final ST construction. Additionally, the ST inversion method has a specific scope of application, considering the marine geographical environment. A single inversion method is difficult to efficiently handle, given multiple types of ST modeling environments. Different inversion methods have their own modeling advantages. In addition, different sea surface gravity field elements

have their advantages in constructing ST models in different marine environments. The details on the applicability and limitations of the different inversion methods are as follows:

(1) The regression analysis (single unit regression or multiple regression) may not be applicable to sea areas with sparse bathymetric data due to the scale factor or grid process of bathymetric data inversion. The modeling effect would be more suited for target sea areas with more bathymetric data and uniform distribution [30]. However, if the whole test area applies one scale factor, regression analysis can be reluctantly applied to ST construction in areas with sparse (or even absent) bathymetric data.

(2) Frequency-domain inversion is often applied to ST inversion in regular rectangular areas [18,19], given that its use for land-sea boundaries does not produce good results.

(3) Although nonlinear iterative least-square inversion in the space domain [26,31] and simulated annealing (SA) inversion [29,32] can be used to recover discrete and irregular bathymetry data in areas without shipborne bathymetric data, their low computational efficiency restricts rapid large-scale ST estimation.

(4) GGM constantly adjusts and optimizes the density difference between the crust and seawater based on check results before obtaining the optimal density difference in the target area. GGM has a better modeling effect for areas with extensive bathymetric data and uniform distribution [33]. This method can also be applied to the ST construction in irregular rectangular sea areas.

(5) According to their respective topographic spectrum information, GA reflects relatively low-frequency information, while VGG reflects relatively high-frequency information. Therefore, the GA inversion is applicable for the deeper portions of the sea, while the VGG is more suited for the shallower parts [34].

*2.2. Seafloor Topography Construction Strategy*

When using sea surface gravity data to build a large-scale ST model, the usual processing method is as follows: a large sea area is divided into many small grids, and the ST data in each grid cell is recovered and spliced to obtain a wide range of ST data [13,35]. For large-scale ST modeling, however, there are many types of ST modeling environments, such as land-sea boundary, full ocean coverage, shallow sea and deep sea. Different gravity data have varying suitability in ST modeling for different marine environments. For example, [34] showed that for sea depths greater than 1500 m, GA offers better inversion effect. For sea depths of less than 1500 m, the ST inversion using VGG is better.

Based on the above analysis, the ST modeling strategy (see Table 1) was developed, considering the efficiency of ST construction, applicability of the inversion method and advantages of sea surface gravity field for different ST modeling environments. Table 1 is described as follows:

(1) "Many" means that the number of bathymetric survey results is greater than 50% of the total bathymetric survey results in the target sea area with $1'$ grid as the reference. "Less" means the amount of bathymetric survey results is less than 50%.

(2) "Grid proportion" (under bathymetric survey results) means that the bathymetric survey results are thinned into $1'$ grid intervals. The number of grids containing bathymetric data accounts for the proportion of the total number of grids.

(3) When the distribution uniformity index of bathymetric data is less than 1.8, the distribution is uniform; otherwise, the distribution is considered uneven [36]. Note that during uniformity evaluation, the bathymetric survey results are diluted at $2'$ intervals.

Note that the ST modeling sea area is irregular when the ST model is constructed in the sea-land boundary zone. The regression analysis is used to determine the scale factor of ST and gravity data for the target sea area. This means that the whole target sea area shares the same scale factor. In contrast, the scale factor is determined from the moving window of the target sea area completely covered by seawater, and the whole target sea area does not share the same scale factor.

**Table 1.** Construction strategy of the ST model.

| Inversion Environment | Average Sea Depth | Bathymetric Survey Results | | Distribution of Bathymetric Data | Inversion Method | Gravity Field Elements |
|---|---|---|---|---|---|---|
| | | Total | Grid Proportion | | | |
| Sea-land boundary area | >1500 m | many/less | >30% | uniformity | GGM | GA |
| | | | | nonuniformity | regression analysis | GA |
| | | | <30% | uniformity/nonuniformity | regression analysis | GA |
| | <1500 m | many/less | >30% | uniformity | GGM | GA |
| | | | | nonuniformity | regression analysis | VGG |
| | | | <30% | uniformity/nonuniformity | regression analysis | VGG |
| Sea Area completely covered by ocean | >1500 m | many | >30% | uniformity | GGM/regression analysis | GA |
| | | | | nonuniformity | Iterative inversion in frequency domain | GA |
| | | | <30% | uniformity/nonuniformity | Iterative inversion in frequency domain | GA |
| | | less | >30% | uniformity | GGM/Iterative inversion in frequency domain | GA |
| | | | | nonuniformity | Iterative inversion in frequency domain | GA |
| | | | <30% | uniformity/nonuniformity | Iterative inversion in frequency domain | GA |
| | <1500 m | many | >30% | uniformity | GGM/regression analysis | GA/VGG |
| | | | | nonuniformity | Iterative inversion in frequency domain | VGG |
| | | | <30% | uniformity/nonuniformity | Iterative inversion in frequency domain | VGG |
| | | less | >30% | uniformity | GGM/Iterative inversion in frequency domain | GA/VGG |
| | | | <30% | uniformity/nonuniformity | Iterative inversion in frequency domain | VGG |

Note: GGM denotes Gravity-Geologic Method. GA denotes gravity anomaly (GA); VGG denotes vertical gravity gradient anomaly.

## 3. Seafloor Topography Construction in the South China Sea

We evaluated the feasibility of the proposed ST modeling strategy, using part of the South China Sea as the research area. In Figure 1, the South China Sea is shown in the red box, with a sea area of roughly about $19^{\circ} \times 15^{\circ}$ ($104^{\circ}$ E $\sim 119^{\circ}$ E, $3^{\circ}$ N $\sim 22^{\circ}$ N).

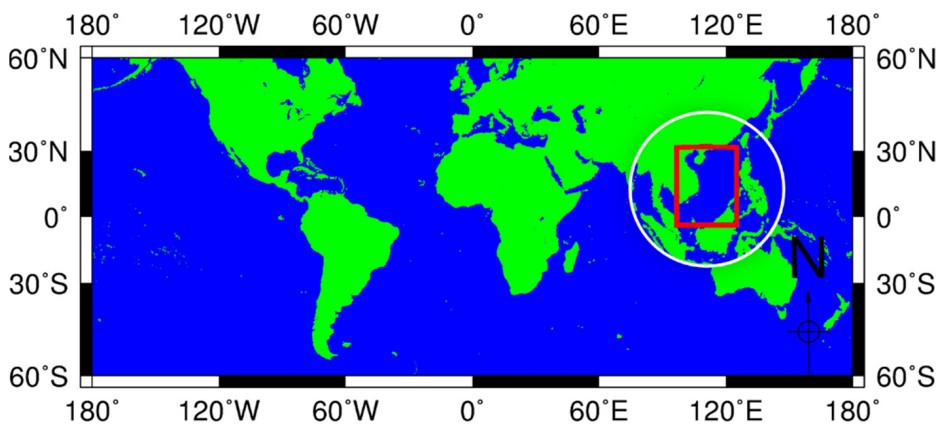

**Figure 1.** The South China Sea Area is shown in the red box.

### 3.1. Data Source and Preprocessing

The data for the ST construction comprises two datasets: bathymetric and sea surface gravity.

#### 3.1.1. Bathymetric Data

The bathymetric dataset included: (1) Multi-source bathymetric survey results released by GEBCO_2019, including bathymetric survey data from single-beam and multi-beam echo-sounders, seismic methods, the ENC and radar sensors. The distribution of these data in the South China Sea is shown in Figure 2a. Using $1'$ grid reference, the coverage area for the bathymetric survey data in Figure 2a accounts for 24.59% of the total study sea area. (2) The single-beam and multi-beam bathymetric data were obtained from the National Geophysical Data Center (NGDC). Since the shipborne bathymetric data released by the NGDC are original survey results prior to data editing for outliers, simple gross error processing was performed on the bathymetric data using S&S V20.1 bathymetry model as an *a priori* model with a $2\sigma$ criterion. Gross error processing method is the same as [19]. The statistical results of the original bathymetry data, processed bathymetry data and the residual are summarized in Table 2. The standard deviation (SD) of bathymetry residual is generally about 419 m. A total of 12,937 suspicious points were removed from the original bathymetry data, accounting for 2.19% of the total dataset. Figure 2b presents the distribution of the bathymetry data after processing. Using $1'$ grid reference, the coverage area of bathymetric data in Figure 2b accounts for 12.60% of the total area.

Comparing Figure 2a,b, the GEBCO_2019 multi-source bathymetric data is significantly greater than the shipborne bathymetric data from the NGDC. In Figure 2c, the red tracks represent the NGDC results, while the black lines present the GEBCO_2019 multi-source bathymetric data distribution. Using $1'$ grid reference, the area covered by the GEBCO_2019 multi-source bathymetric data is about 12% larger than the area covered by the NGDC data.

To evaluate the quality of the constructed ST models, the processed NGDC bathymetric survey data were randomly and uniformly divided into two groups, about 2/3 (389,535) were control points, and about 1/3 (188,432) were checkpoints. Control points participated in ST modeling, and checkpoints were used as independent data to evaluate ST results. The distribution of control points and checkpoints is shown in Figure 3a; the red dots represent the control points and the black dots represent the checkpoints.

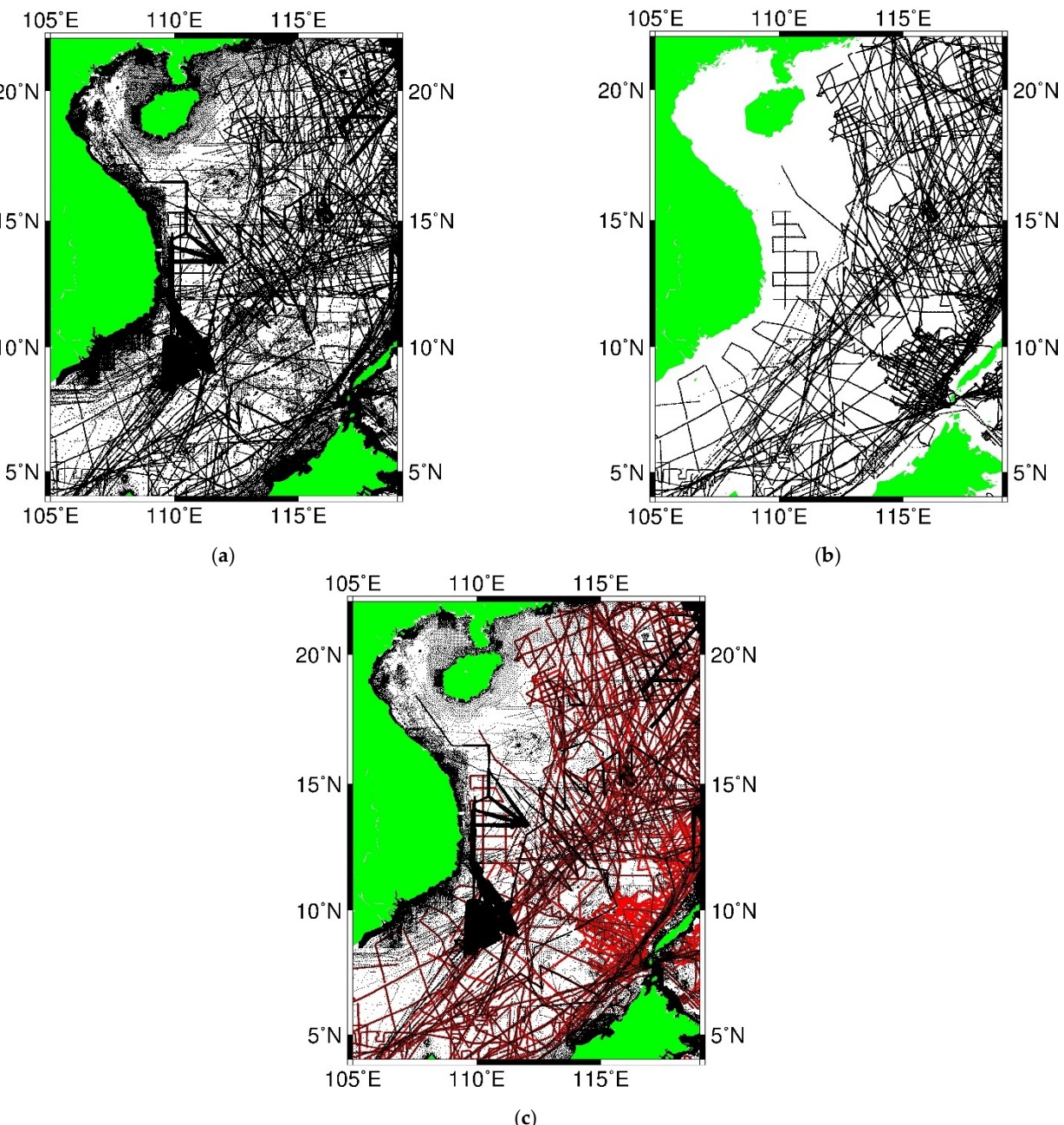

**Figure 2.** Distribution of bathymetric data. (**a**) The black dots represnt bathymetric data from GEBCO_2019. (**b**) The black dots represnt bathymetric data from NGDC. (**c**) Distribution comparison between NGDC and GEBCO_2019, where the black dots represent data from GEBCO_2019 and the red dots represent data from NGDC.

**Table 2.** Statistical results of NGDC bathymetric data (unit: m).

| Datatype | Number | Max. | Min. | Mean | SD |
|---|---|---|---|---|---|
| Raw bathymetric data | 590,904 | 0.00 | −10,251.50 | −2229.43 | 1305.01 |
| Bathymetry residual | 590,904 | 4704.52 | −10,160.06 | −5.87 | 418.88 |
| Processed bathymetric data | 577,967 | 0.00 | −4997.00 | −2250.61 | 1301.20 |

Note: SD denotes the standard deviation.

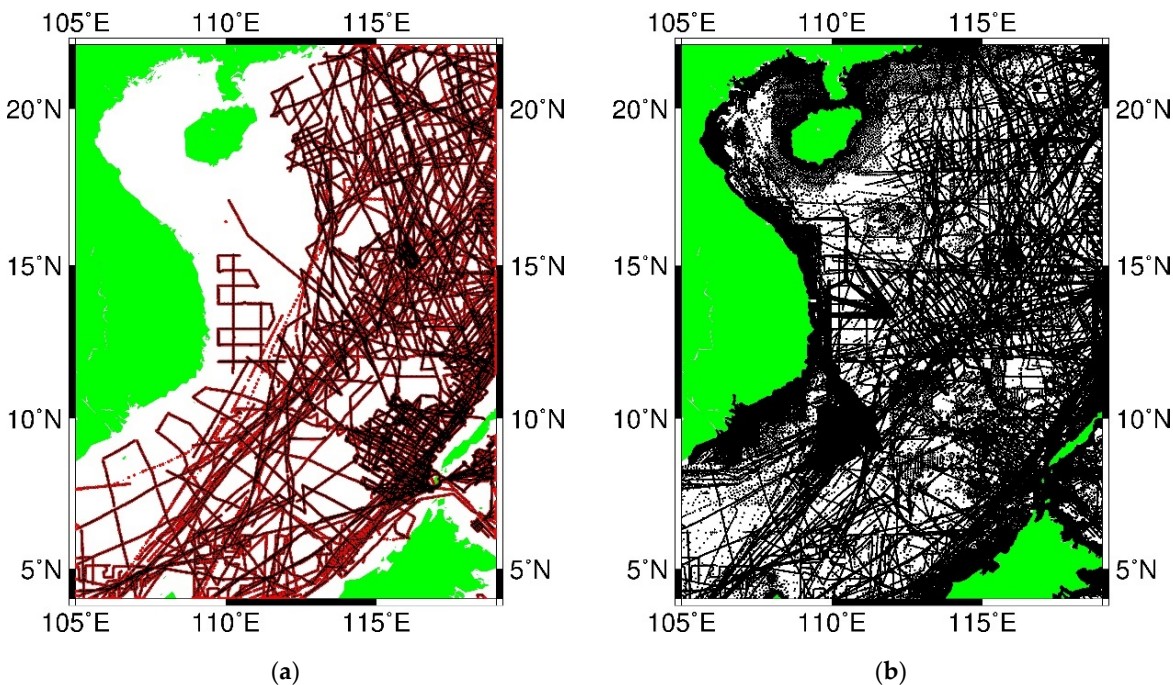

**Figure 3.** Distribution of shipborn depth data. (**a**) Distribution of control points and checkpoints; the red dots represent the control points and the black dots represent the checkpoints. (**b**) The black dots represent fusion results of control point and GEBCO_2019.

As shown in Figure 2, there were a lot of overlapping areas between the NGDC shipborne and the GEBCO_2019 multi-source bathymetry data. To fully utilize the measurement results from the different platforms, we used the fusion method to deal with the results at repeated cross measuring points. The integration strategy was implemented using the following procedure: (1) The South China Sea Area was divided into 15″ grids; and (2) The average sea depth at each grid cell was calculated from the different sources, and was then used as the new cell value. The fusion results for the control points and GEBCO_2019 multi-source bathymetric data are shown in Figure 3b. The fusion bathymetric data account for 27.30% of the total area, with a 1′ grid reference.

The sea depths at control points, the GEBCO_2019 multi-source sea depths and the fusion results of the two datasets, are shown in Table 3. As shown in the table, the shallowest sea depth in the South China Sea is close to zero, while the deepest is close to 5000 m. The average sea depth is about 2100–2200 m, with standard deviation of about 1300 m. The values suggest that the ST in the study area is complex and highly fluctuating, and includes both shallow and deep-sea regions.

**Table 3.** Statistics of sea depth in the South China Sea (unit: m).

| Datatype | Max. | Min. | Mean | SD |
|---|---|---|---|---|
| Sea depth in control points | 0.00 | −4997.00 | −2246.80 | 1301.47 |
| GEBCO_2019 multi-source sea depth | 0.00 | −4979.00 | −2107.45 | 1312.29 |
| Fusion results of depth of control points and GEBCO_2019 | 0.00 | −4986.00 | −2108.16 | 1300.73 |

Note: SD denotes the standard deviation.

### 3.1.2. Sea Surface Gravity Data

Sea surface gravity data includes sea surface GA and VGG from version V29.1 released by the UCSD SIO (Scripps Institution of Oceanography, University of California San Diego) on 22 November 2019. The sea surface GA and VGG are referred to as GA_29.1 and VGG_29.1, and their distributions are presented in Figure 4a,b. The statistical results

in Table 4 show that the maximum, average and SD for sea surface GA are 209 mGal, 5 mGal and 23 mGal, respectively, while for VGG, the maximum, minimum and SD are 505.99 Eotvos, −656.53 Eotvos and 20.54 Eotvos.

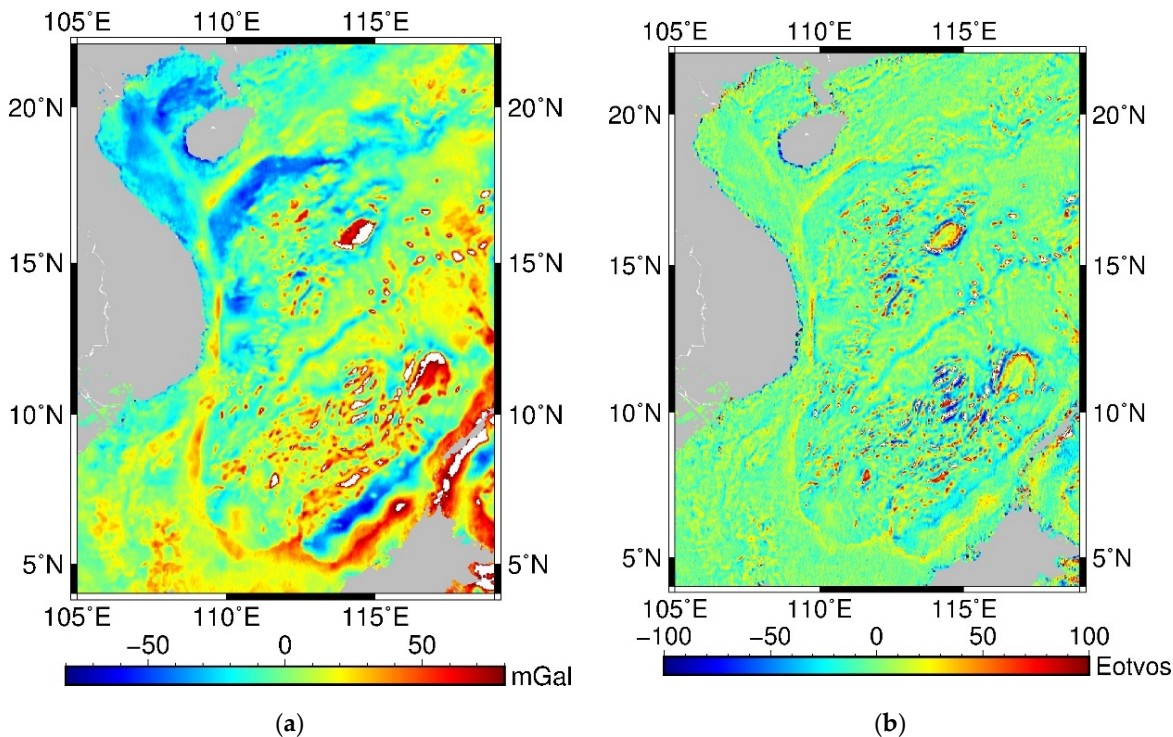

**Figure 4.** Sea surface gravity data in the South China Sea. (**a**) Gravity anomaly, called GA_29.1; (**b**) Vertical gravity gradient anomaly, called VGG_29.1.

**Table 4.** Statistical results of sea surface gravity data in the South China Sea.

| Datatype | Max. | Min. | Mean | SD |
|---|---|---|---|---|
| GA (mGal) | 209.79 | −170.95 | 5.00 | 23.82 |
| VGG (Eotvos) | 505.99 | −656.53 | −0.35 | 20.54 |

Note: SD denotes the standard deviation; GA denotes gravity anomaly; and VGG denotes vertical gravity gradient anomaly.

### 3.2. Seafloor Topography Model Construction

The specific operation steps for the ST construction are as follows:

(1) The South China Sea was segmented using a 2° square grid.
(2) Following the ST model construction strategy in Table 1, we recovered the ST of the sea area with a side length of 2° one by one.
(3) The 2° × 2° regional ST data were spliced to obtain the ST data set.

Bathymetric survey data (obtained by the fusion processing method), the GA and the VGG were used as input data for the ST construction. The ST model with a 1′ resolution was constructed by combining multiple gravity data and integrating multi-class inversion methods (see Figure 5a). The ST model is hereafter referred to as the BAT_ SCS model.

Based on the statistical results, the integrated bathymetric data only accounts for about 27% of the study area using the 1′ grid reference. This means that nearly three-quarters of the ST in the South China Sea have been estimated using gravity data, and the results are summarized in Figure 5b. The SIO V20.1, DTU18 and ETOPO1 ST models were then generated, and the results are shown in Figure 6a–c. The statistical results of the four ST models are shown in Table 5.

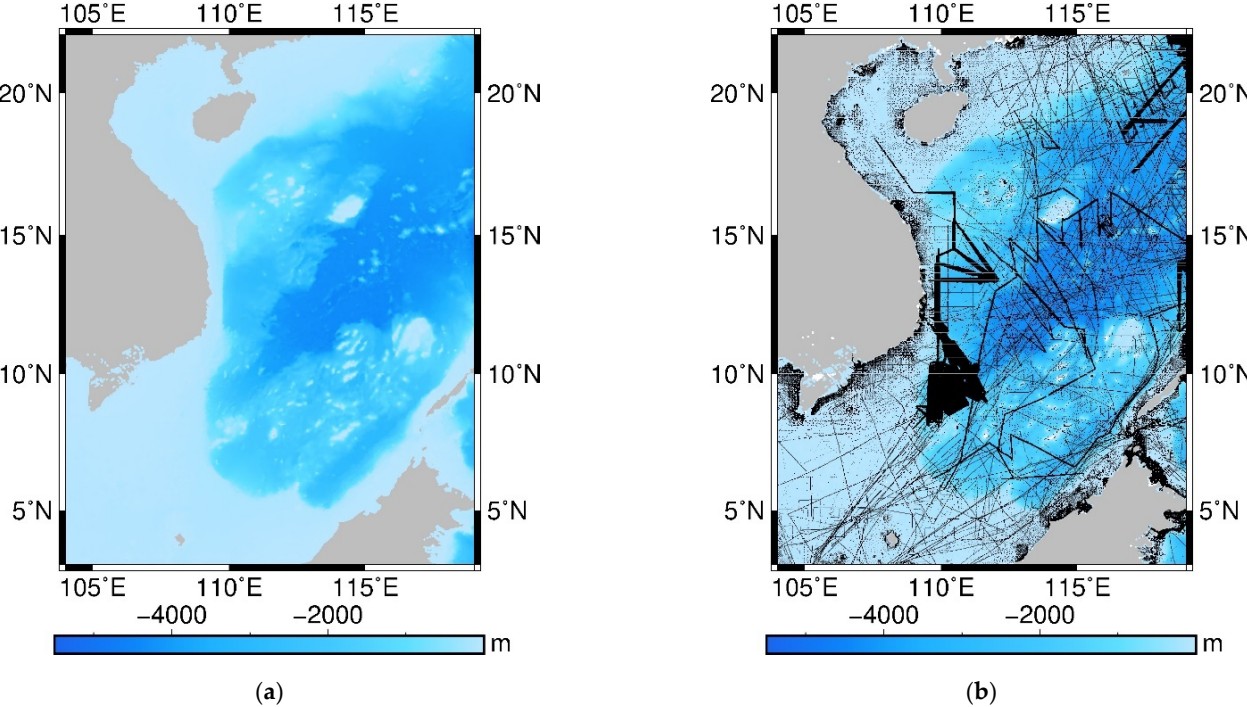

(**a**)
(**b**)

**Figure 5.** ST models. (**a**) BAT_SCS model. (**b**) The colored part in the figure represents ST derived from gravity data, and the black dots represent the shipborne bathymetric data.

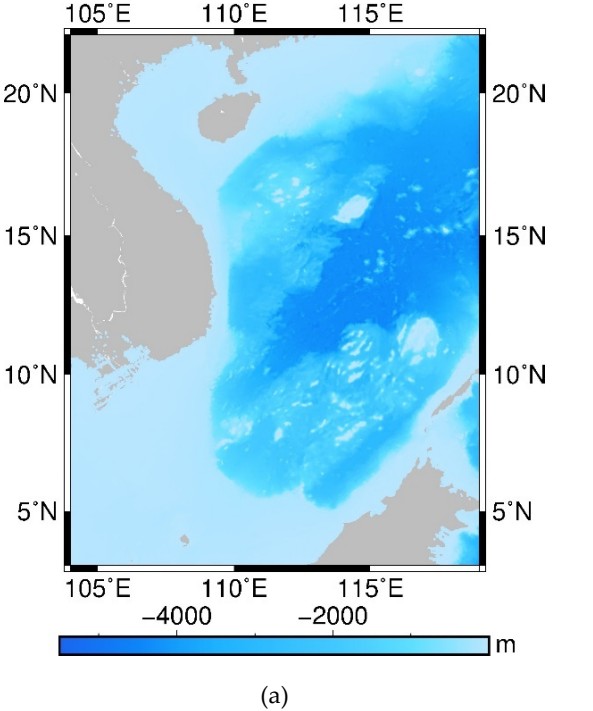
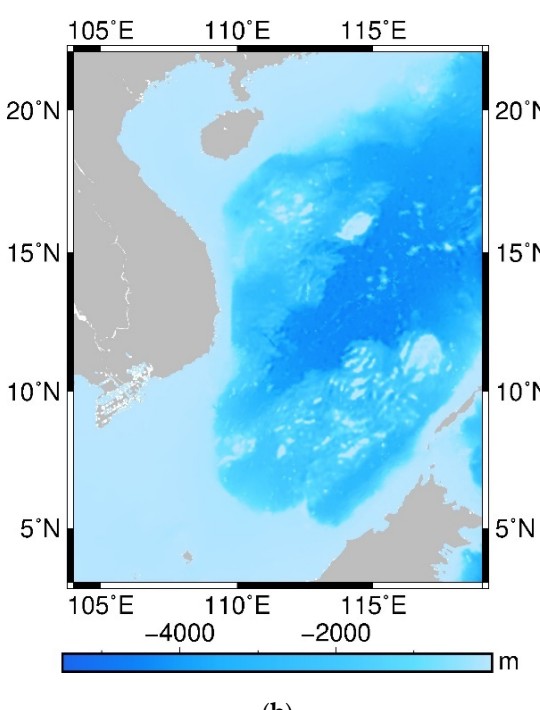

(a)
(**b**)

**Figure 6.** *Cont*.

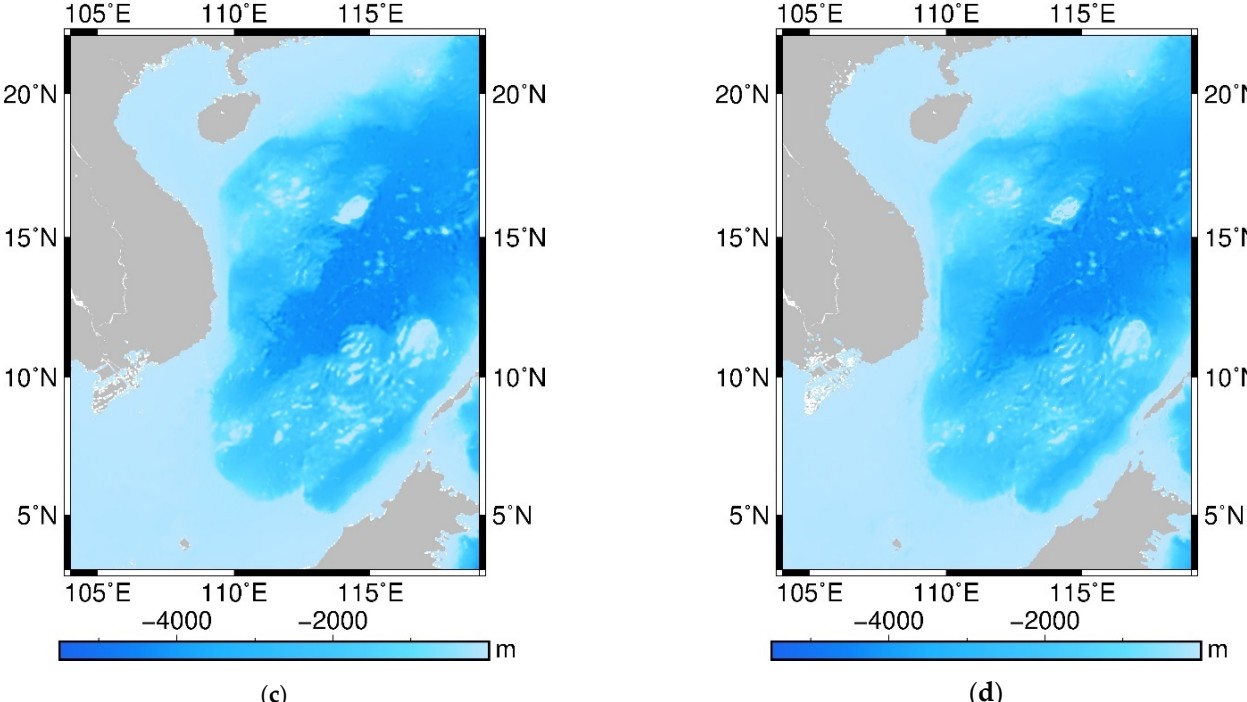

**Figure 6.** ST models. (**a**) SIO V20.1 bathymetry model; (**b**) DTU18 bathymetry model; (**c**) ETOPO1 bathymetry model; and (**d**) BAT_VGG bathymetry model.

**Table 5.** Statistics of sea depth model (unit: m).

| ST Model | Max. | Min. | Mean | SD |
|----------|------|------|------|-----|
| BAT_SCS | 0.00 | −4956.00 | −1428.75 | 1516.78 |
| SIO V20.1 | 0.00 | −4978.00 | −1423.49 | 1519.04 |
| DTU18 | 0.00 | −4959.00 | −1418.73 | 1523.71 |
| ETOPO1 | −1.00 | −4971.00 | −1420.06 | 1526.57 |
| BAT_VGG | 0.00 | −4804.00 | −1398.21 | 1486.48 |

Note: SD denotes the standard deviation.

Comparing Figures 5a and 6, the BAT_SCS model constructed in the paper is highly consistent with the SIO V20.1 model, the DTU18 model and the ETOPO1 model of the South China Sea. The statistical results in Table 5 show that the maximum, minimum, mean and SD for sea depth in the BAT_SCS model are also comparable to the SIO V20.1, DTU18 and ETOPO1 models. The results suggest that the data processing method for the ST modeling is correct, and that the calculation results are highly reliable.

### 3.3. Accuracy Evaluation of Seabed Terrain Model

Two sea areas in the South China Sea (deep sea area and shallow sea area) were selected and used for quality evaluation, and the scope of verification area A is $5^{\circ} \times 5^{\circ}$ ($110^{\circ}$ E $\sim 115^{\circ}$ E, $10^{\circ}$ N $\sim 15^{\circ}$N). The distribution of checkpoints in the verification sea area is shown in Figure 7a, with a total of 19,118 bathymetric points. The verification area B is $3^{\circ} \times 5^{\circ}$ ($105^{\circ}$ E $\sim 110^{\circ}$ E, $5^{\circ}$ N $\sim 8^{\circ}$ N), and checkpoint distribution is presented in Figure 7b, with a total of 2355 measurement points. Table 6 summarizes the statistical results of the checkpoints for Areas A and B. The average sea depth for Area A is about 3800 m, and it is 200 m for Area B; the shallowest part is only 24 m.

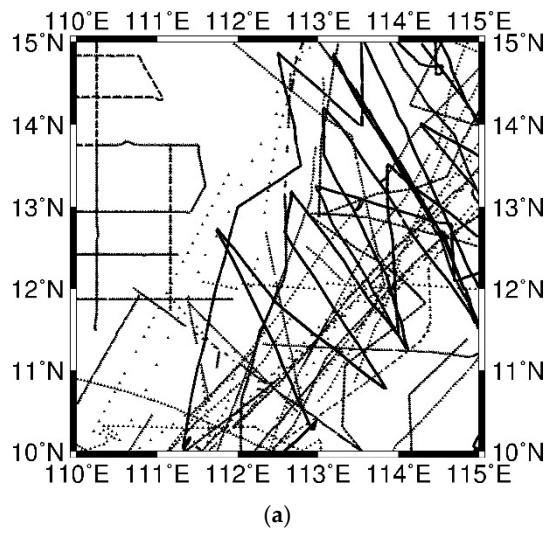
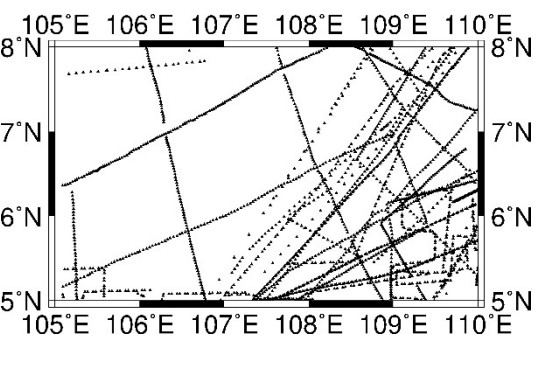

(**a**)

(**b**)

**Figure 7.** Distribution of checkpoints in the test areas. (**a**) Verification Sea Area A; (**b**) Verification Sea Area B.

**Table 6.** The statistics of checkpoints in verification sea area (unit: m).

| Sea Area | Number | Max. | Min. | Mean | SD |
|---|---|---|---|---|---|
| Sea Area A | 19,118 | −322.00 | −4618.00 | −3818.47 | 789.61 |
| Sea Area B | 2355 | −24.00 | −1936.00 | −197.15 | 276.62 |

Note: SD denotes the standard deviation.

Using the actual independent bathymetric survey results for Sea Areas 1 and 2 as the external verification reference, the ST model value was interpolated by the bilinear interpolation method. The interpolation results were then compared to the actual bathymetric values at the checkpoints. The final verification statistical results for the ST models are presented in Tables 7 and 8. In the tables, "primary checking (PC)" indicates the statistical results for all checkpoints, while "secondary checking (SC)" represents the results after eliminating suspicious points according to the 3σ criterion. For example, 98.15% of the checkpoints in Area A were retained for the secondary check of the BAT_SCS model, which was also called the data retention rate (DRR). The relative accuracy (RA) indicates the ratio of the SD of the check results to the average sea depth of the checkpoints.

**Table 7.** Statistical check results of ST models in Sea Area A (unit: m).

| ST Model | Max. | Min. | Mean | RMS | SD | CC | RA | DRR | Remarks |
|---|---|---|---|---|---|---|---|---|---|
| BAT_SCS | 1125.47 | −1174.22 | 2.99 | 84.81 | 84.76 | 0.9942 | 2.22% | 100% | PC |
|  | 256.61 | −249.97 | 2.43 | 51.09 | 51.04 | 0.9978 | 1.33% | 98.15% | SC |
| SIO V20.1 | 970.07 | −1521.07 | 1.72 | 87.52 | 87.50 | 0.9938 | 2.29% | 100% | PC |
|  | 263.99 | −260.57 | 0.68 | 59.46 | 59.45 | 0.9971 | 1.55% | 97.92% | SC |
| DTU18 | 895.76 | −1436.69 | −0.20 | 90.67 | 90.67 | 0.9934 | 2.37% | 100% | PC |
|  | 271.78 | −271.68 | 0.09 | 61.91 | 61.91 | 0.9968 | 1.61% | 97.85% | SC |
| ETOPO1 | 1598.38 | −1467.41 | −7.57 | 190.14 | 190.00 | 0.9720 | 4.98% | 100% | PC |
|  | 559.75 | −575.78 | −2.18 | 135.87 | 135.85 | 0.9848 | 3.53% | 97.04% | SC |

Note: ST denotes seafloor topography; SD denotes standard deviation; CC denotes correlation coefficient; RA denotes relative accuracy; DRR denotes data retention rate; PC denotes primary checking; and SC denotes secondary checking.

**Table 8.** Statistical check results of ST model in Sea Area B (unit: m).

| ST model | Max. | Min. | Mean | RMS | SD | CC | RA | DRR | Remarks |
|---|---|---|---|---|---|---|---|---|---|
| BAT_SCS | 577.85 | −497.95 | −3.22 | 33.95 | 33.80 | 0.9927 | 17.15% | 100% | PC |
| | 97.34 | −102.12 | −2.37 | 15.92 | 15.75 | 0.9983 | 8.23% | 98.90% | SC |
| SIO V20.1 | 424.45 | −449.40 | −2.14 | 26.73 | 26.65 | 0.9955 | 13.52% | 100% | PC |
| | 71.94 | −82.03 | −1.09 | 11.43 | 11.38 | 0.9992 | 6.00% | 98.39% | SC |
| DTU18 | 411.98 | −746.18 | −2.39 | 35.06 | 34.99 | 0.9921 | 17.75% | 100% | PC |
| | 102.24 | −105.89 | −1.39 | 12.49 | 12.41 | 0.9990 | 6.46% | 98.98% | SC |
| ETOPO1 | 351.96 | −1053.14 | −7.66 | 50.35 | 49.78 | 0.9843 | 25.25% | 100% | PC |
| | 135.95 | −156.55 | −3.37 | 20.78 | 20.51 | 0.9973 | 10.96% | 98.09% | SC |

Note: ST denotes seafloor topography; RMS denotes root mean square; SD denotes standard deviation; CC denotes correlation coefficient; RA denotes relative accuracy; DRR denotes data retention rate; PC denotes primary checking; and SC denotes secondary checking.

The SD for the BAT_SCS model in Sea Area A (deep-sea area) was 51.04 m, which was comparatively better than those for the SIO V20.1 and DTU18 models. Compared with the ETOPO1 model, the BAT_SCS model accuracy was more than two times higher. For Sea Area B (see Table 8), the BAT_SCS model accuracy was comparable to the SIO V20.1 and DTU18 models, and significantly better by nearly 30% than the ETOPO1 model. The root mean square (RMS) for the BAT_SCS model in Sea Area B (shallow-sea area) was 15.92 m. The results suggest that the ST model construction strategy proposed in this study applies to multiple marine environments, including sea-land boundaries and open seas, and that the ST construction results are of high quality.

## 4. Construction and Accuracy Evaluation of Global Seafloor Topography Model

### 4.1. Global Seafloor Topography Model Construction

The ST extent constructed using satellite altimetry gravity data was (180° W ∼ 180° N, 60° S ∼ 64° N). The gravity data released by SIO V29.1 for GA and VGG are shown in Figures 8 and 9, and the bathymetric data distribution is presented in Figure 10. The International Bathymetric Chart of the Southern Ocean (IBCSO) could be used to supplement ST data in the range of (180° W ∼ 180° N, 60° S ∼ 90° S) [37], while the International Bathymetric Chart of the Arctic Ocean (IBCAO) could supplement ST data in the range of (180° W ∼ 180° N, 64° N ∼ 90° N) [38]. Finally, the global ST model was constructed following the strategy in Table 1 using SIO V29.1 GA and VGG (Figure 11), and is hereafter referred to as the STO_IEU2020 model.

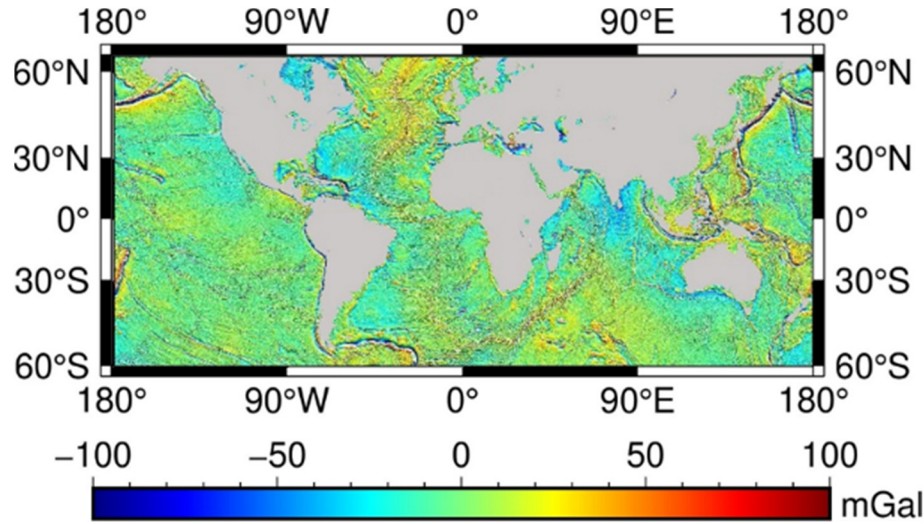

**Figure 8.** SIO V29.1 gravity anomaly.

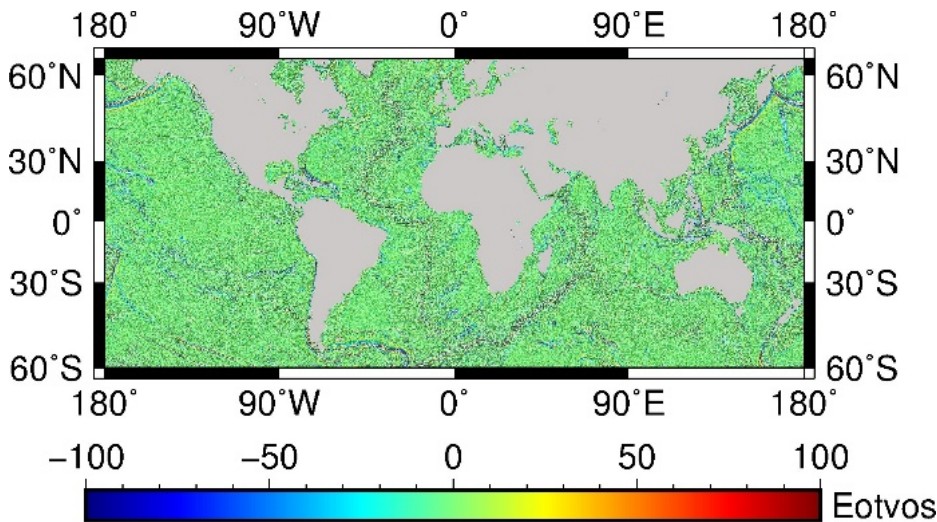

**Figure 9.** SIO V29.1 vertical gravity gradient anomaly.

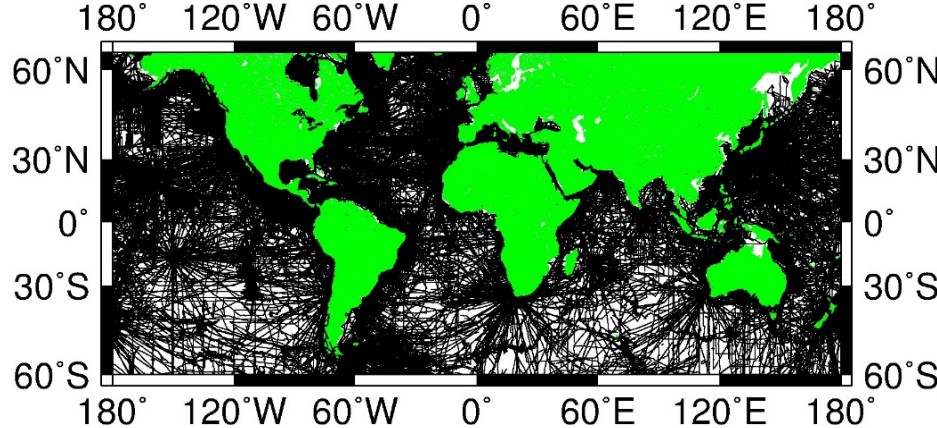

**Figure 10.** The distribution of shipborne depth data.

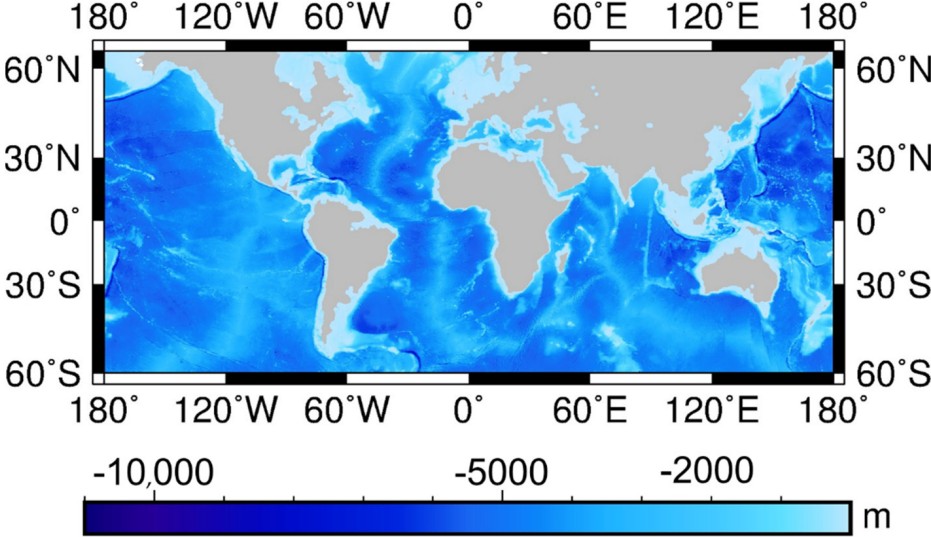

**Figure 11.** STO_IEU2020 global ST model.

### 4.2. Accuracy Evaluation of Seafloor Topography Model

To evaluate the quality of the STO_IEU2020 model, five sea areas in the Atlantic Ocean, Indian Ocean and the Pacific Ocean were selected, as shown in Figure 12. The extent of the evaluation areas are as follows: Sea Area 1, $30^{\circ} \times 30^{\circ}$ ($20^{\circ}$N $\sim 50^{\circ}$N, $20^{\circ}$W $\sim 50^{\circ}$W); Sea Area 2, $55^{\circ} \times 25^{\circ}$ ($10^{\circ}$S $\sim 45^{\circ}$S, $5^{\circ}$W $\sim 30^{\circ}$W); Sea Area 3, $40^{\circ} \times 15^{\circ}$ ($30^{\circ}$S $\sim 10^{\circ}$N, $60^{\circ}$E $\sim 75^{\circ}$E); Sea Area 4, $30^{\circ} \times 20^{\circ}$ ($10^{\circ}$N $\sim 40^{\circ}$N, $150^{\circ}$E $\sim 170^{\circ}$E); and Sea Area 5, $55^{\circ} \times 30^{\circ}$ ($45^{\circ}$S $\sim 10^{\circ}$N, $100^{\circ}$W $\sim 130^{\circ}$W). Figure 13 shows the distribution of ship-borne sea depth used as checkpoints in the evaluation areas, and the statistical results are summarized in Table 9.

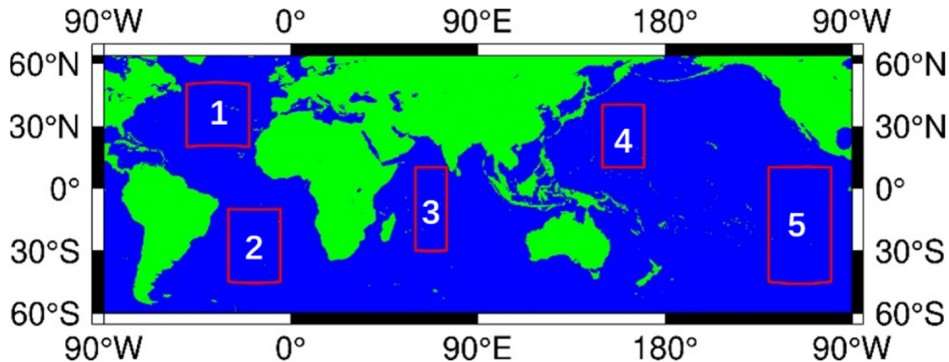

**Figure 12.** Schematic diagram of assessing sea areas.

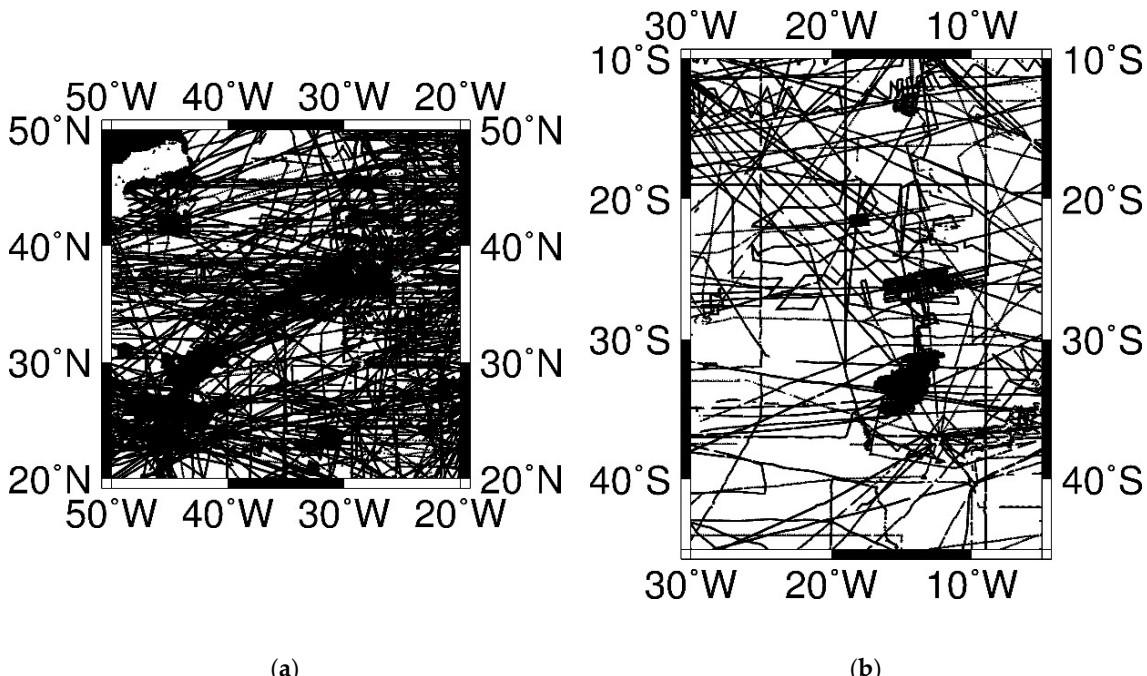

(**a**)  (**b**)

**Figure 13.** *Cont.*

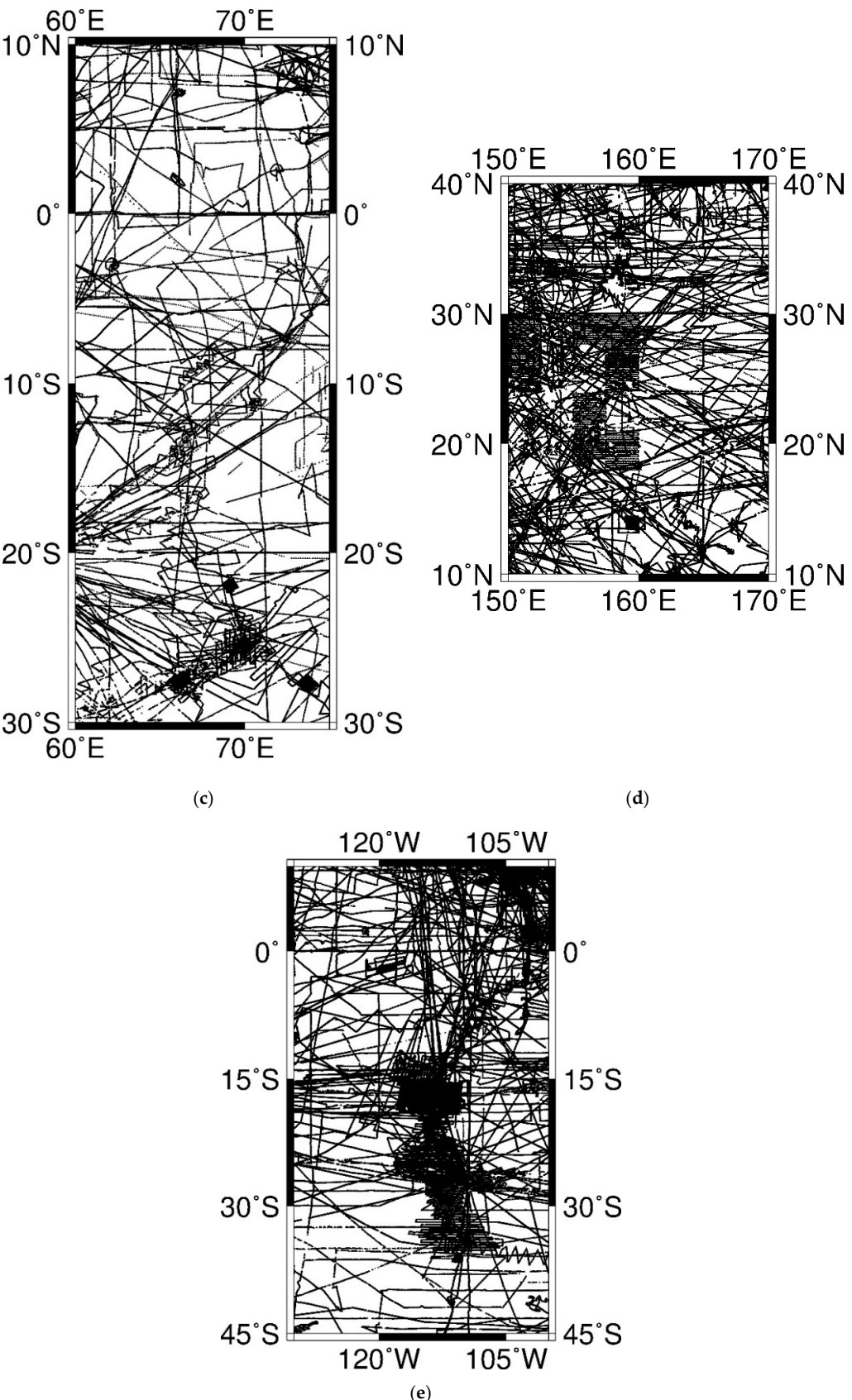

**Figure 13.** Distribution of checkpoints in the sea area. (**a**) Sea Area 1; (**b**) Sea Area 2; (**c**) Sea Area 3; (**d**) Sea Area 4; and (**e**) Sea Area 5.

**Table 9.** Statistical results of checkpoints in evaluation area (unit: m).

| Sea Area | Number | Max. | Min. | Mean | SD |
|---|---|---|---|---|---|
| Sea Area 1 | 577,354 | −27.70 | −7048.00 | −3548.38 | 1152.65 |
| Sea Area 2 | 187,400 | −113.40 | −6378.00 | −3929.96 | 868.89 |
| Sea Area 3 | 132,650 | −1.00 | −6369.70 | −3463.46 | 1033.02 |
| Sea Area 4 | 168,163 | −30.00 | −7551.00 | −5170.74 | 1052.01 |
| Sea Area 5 | 642,483 | −329.00 | −5567.00 | −3321.48 | 546.54 |

Note: SD denotes standard deviation.

As shown in Table 9, a large number of checkpoints were used in evaluating each sea area: 577,354 for Sea Area 1; 187,400 for Sea Area 2; 132,650 for Sea Area 3; 168,163 for Sea Area 4; and 642,483 for Sea Area 5. Having high amounts of samples would ensure the reliability of statistical results. The statistical results for the STO_IEU2020 model, SIO V20.1 model, DTU18 model and ETOPO1 model in each sea area are summarized in Table 10.

**Table 10.** Statistical checking results of bathymetry models (Unit: m).

| Sea Area | ST Model | Max. | Min. | Mean | RMS | SD | CC | RA | DRR | Remarks |
|---|---|---|---|---|---|---|---|---|---|---|
| Sea Area 1 | STO_IEU2020 | 3268.71 | −4377.91 | 7.25 | 189.08 | 188.94 | 0.9865 | 5.32% | 100% | PC |
| | | 573.93 | −558.99 | 1.41 | 100.26 | 100.26 | 0.9961 | 2.81% | 97.88% | SC |
| | SIO V20.1 | 3319.09 | −4570.82 | 12.52 | 199.38 | 198.99 | 0.9850 | 5.61% | 100% | PC |
| | | 609.40 | −584.44 | 4.22 | 94.89 | 94.79 | 0.9965 | 2.65% | 97.78% | SC |
| | DTU18 | 3219.56 | −4573.67 | 9.97 | 203.98 | 203.74 | 0.9843 | 5.74% | 100% | PC |
| | | 621.18 | −601.18 | 1.71 | 102.13 | 102.12 | 0.9960 | 2.86% | 97.77% | SC |
| | ETOPO1 | 3339.18 | −4625.06 | 8.78 | 250.86 | 250.71 | 0.9761 | 7.07% | 100% | PC |
| | | 760.70 | −743.28 | 2.05 | 178.75 | 178.74 | 0.9877 | 5.01% | 97.86% | SC |
| Sea Area 2 | STO_IEU2020 | 1853.36 | −1542.87 | 6.77 | 79.70 | 79.41 | 0.9958 | 2.02% | 100% | PC |
| | | 245.02 | −231.44 | 4.98 | 63.82 | 63.63 | 0.9973 | 1.62% | 98.44% | SC |
| | SIO V20.1 | 2095.37 | −1804.35 | 6.98 | 69.59 | 69.24 | 0.9968 | 1.76% | 100% | PC |
| | | 214.69 | −200.71 | 5.45 | 51.78 | 51.49 | 0.9983 | 1.31% | 98.39% | SC |
| | DTU18 | 2193.93 | −1576.47 | 6.58 | 88.89 | 88.64 | 0.9948 | 2.26% | 100% | PC |
| | | 272.45 | −259.34 | 4.66 | 62.10 | 61.92 | 0.9975 | 1.58% | 98.06% | SC |
| | ETOPO1 | 1991.41 | −3119.93 | −10.65 | 191.86 | 191.56 | 0.9750 | 4.90% | 100% | PC |
| | | 563.90 | −585.04 | −5.15 | 152.42 | 152.33 | 0.9841 | 3.89% | 98.06% | SC |
| Sea Area 3 | STO_IEU2020 | 2591.86 | −2384.37 | 1.42 | 132.09 | 132.09 | 0.9918 | 3.81% | 100% | PC |
| | | 397.66 | −394.77 | 0.24 | 105.97 | 105.97 | 0.9947 | 3.06% | 98.47% | SC |
| | SIO V20.1 | 2664.39 | −2270.22 | −19.62 | 159.80 | 158.59 | 0.9882 | 4.58% | 100% | PC |
| | | 456.09 | −495.36 | −12.78 | 122.49 | 121.82 | 0.9931 | 3.52% | 97.76% | SC |
| | DTU18 | 2749.27 | −2307.27 | −19.54 | 177.08 | 176.00 | 0.9854 | 5.08% | 100% | PC |
| | | 508.32 | −547.44 | −9.42 | 129.88 | 129.54 | 0.9922 | 3.75% | 97.42% | SC |
| | ETOPO1 | 3111.41 | −2538.04 | −26.13 | 238.87 | 237.44 | 0.9733 | 6.86% | 100% | PC |
| | | 685.98 | −738.37 | −24.83 | 210.92 | 209.45 | 0.9790 | 6.04% | 98.37% | SC |
| Sea Area 4 | STO_IEU2020 | 2476.39 | −5055.21 | 1.86 | 108.75 | 108.74 | 0.9946 | 2.10% | 100% | PC |
| | | 327.82 | −324.33 | −0.84 | 56.48 | 56.47 | 0.9985 | 1.09% | 98.54% | SC |
| | SIO V20.1 | 2450.31 | −5079.75 | 1.66 | 120.59 | 120.58 | 0.9934 | 2.33% | 100% | PC |
| | | 363.12 | −359.59 | 0.77 | 50.04 | 50.03 | 0.9988 | 0.96% | 98.55 | SC |
| | DTU18 | 2995.38 | −5106.11 | 3.28 | 133.15 | 133.11 | 0.9920 | 2.57% | 100% | PC |
| | | 402.58 | −395.75 | 0.64 | 52.46 | 52.46 | 0.9987 | 1.01% | 98.58% | SC |
| | ETOPO1 | 2823.45 | −5132.19 | 1.94 | 181.72 | 181.71 | 0.9851 | 3.51% | 100% | PC |
| | | 547.00 | −543.00 | −0.33 | 110.79 | 110.79 | 0.9940 | 2.13% | 97.43% | SC |
| Sea Area 5 | STO_IEU2020 | 1747.17 | −1885.83 | 12.31 | 65.41 | 64.24 | 0.9932 | 1.93% | 100% | PC |
| | | 205.04 | −180.42 | 11.23 | 50.66 | 49.40 | 0.9960 | 1.49% | 98.31 | SC |
| | SIO V20.1 | 1864.49 | −2070.88 | 14.55 | 66.15 | 64.53 | 0.9931 | 1.94% | 100% | PC |
| | | 208.14 | −179.02 | 13.00 | 49.00 | 47.25 | 0.9963 | 1.42% | 98.24% | SC |
| | DTU18 | 1806.71 | −2003.33 | 10.01 | 68.29 | 67.55 | 0.9924 | 2.03% | 100% | PC |
| | | 212.63 | −192.63 | 8.63 | 47.83 | 47.05 | 0.9963 | 1.42% | 98.44% | SC |

**Table 10.** *Cont.*

| Sea Area | ST Model | Max. | Min. | Mean | RMS | SD | CC | RA | DRR | Remarks |
|---|---|---|---|---|---|---|---|---|---|---|
| | ETOPO1 | 2384.05 | −2097.07 | 14.70 | 162.39 | 161.72 | 0.9561 | 4.86% | 100% | PC |
| | | 499.83 | −470.41 | 9.13 | 105.72 | 105.32 | 0.9812 | 3.16% | 98.11% | SC |

Note: ST denotes seafloor topography; SD denotes standard deviation; CC denotes correlation coefficient; RA denotes relative accuracy; DRR denotes data retention rate; PC denotes primary checking; and SC denotes secondary checking.

In the table, the DRR of SC results in the five evaluation sea areas are higher than 97%. For Sea Areas 1 and 2, located in the North Atlantic and South Atlantic, the CC between the STO_IEU2020 model and shipborne measured sea depth are higher than 0.99, with an RA greater than 3%. In Sea Area 1, the SD of the SC for the SIO V20.1, DTU18 and ETOPO1 models were 94.79 m, 102.12 m and 178.74 m, respectively. The SD for the STO_IEU2020 model was 100.26 m. The STO_IEU2020 had slightly lower accuracy than the SIO V20.1 model, comparable to the DTU18 model and much higher than the ETOPO1 model (about 78% improvement). In Sea Area 2, the SD of the SC for the SIO V20.1, DTU18 and ETOPO1 models were 51.49 m, 61.92 m and 152.33 m, respectively, while the accuracy for the STO_IEU2020 model was 63.63 m. The results suggest that for Sea Area 2, the STO_IEU2020 model has comparable accuracy to the SIO V20.1 and DTU18 models, and is significantly better than the ETOPO1 model. For Sea Area 3, the STO_IEU2020 model has an accuracy of 105.97 m, which is slightly better than the SIO V20.1 (122.49 m) and DTU18 (129.54 m) models. Overall, the STO_IEU2020 model strongly correlates with the shipborne sea depths, with CC greater than 0.99 in the five assessed sea areas. The STO_IEU2020 model has similar accuracy to the SIO V20.1 and DTU18 models, and is much better than the ETOPO1 model. The results show that the STO_IEU2020 model performs well, and that the global ST model proposed in this study produces reliable results.

Figure 14 presents the number of checkpoints of the STO_IEU2020, SIO V20.1, DTU18 and ETOPO1 models in the five evaluated sea areas. Most of the differences are concentrated in the ±200 m range. For the STO_IEU2020 model, the checkpoints within ± 100 m checking difference accounted for 78.57%, 87.87%, 72.27%, 92.47% and 93.25% of the total checkpoints in the five sea areas. Checkpoints within the small value range are more concentrated in the STO_IEU2020, SIO V20.1 and DTU18 models compared with the ETOPO1 model. The STO_IEU2020, SIO V20.1 and DTU18 models have comparable numbers of checkpoints at the various difference ranges. In Sea Areas 1, 4 and 5, the curves of the three ST models almost coincide. In Sea Areas 3 and 4, the STO_IEU2020 model is slightly better than the SIO V20.1 and DTU18 models, while in Sea Areas 2 and 5, the SIO V20.1 model performed better than the STO_IEU2020 and DTU18 models. In general, the SIO V20.1 and DTU18 models produced better accuracy results than the ETOPO1 model, and the SIO V20.1 model was slightly better than the DTU18 model.

The STO_IEU2020 model was then evaluated using the SIO V20.1 model as reference. The statistical results between the two ST models are summarized in Table 11. In the table, the CC between the two models for Sea Areas 1 to 5 were over 0.99. After error processing, the DRR was about 98% and the mean difference value was less than 6 m. Except for Sea Area 3, the SD between the two ST models was less than 100 m. In Sea Area 3, the SD was 131.07 m.

To compare the difference between STO_IEU2020 model and SIO V20.1 model, the difference histograms of the two models were calculated in five sea areas (Figure 15). As shown in Figure 15, the differences between the SIO V20.1 and STO_IEU2020 models were evenly distributed on both sides of the zero value, resulting in normal distribution. About 95.62%, 94.29%, 87.39%, 95.59% and 97.90% of the total sea depth points in the five sea areas were within the ±200 m difference range. Results presented in Table 11 and Figure 15 indicate that the STO_IEU2020 and SIO V20.1 models have a high degree of consistency. STO_IEU2020 and SIO V20.1 are so related, we think the reason may be the input data for constructing the STO_IEU2020 model is from SIO V29.1 data source, so we think that

the STO_IEU2020 model agrees well with the SIO V20.1. From another perspective, the findings suggest that the method for global ST model construction proposed in this study is feasible, and that the STO_IEU2020 model is of high quality.

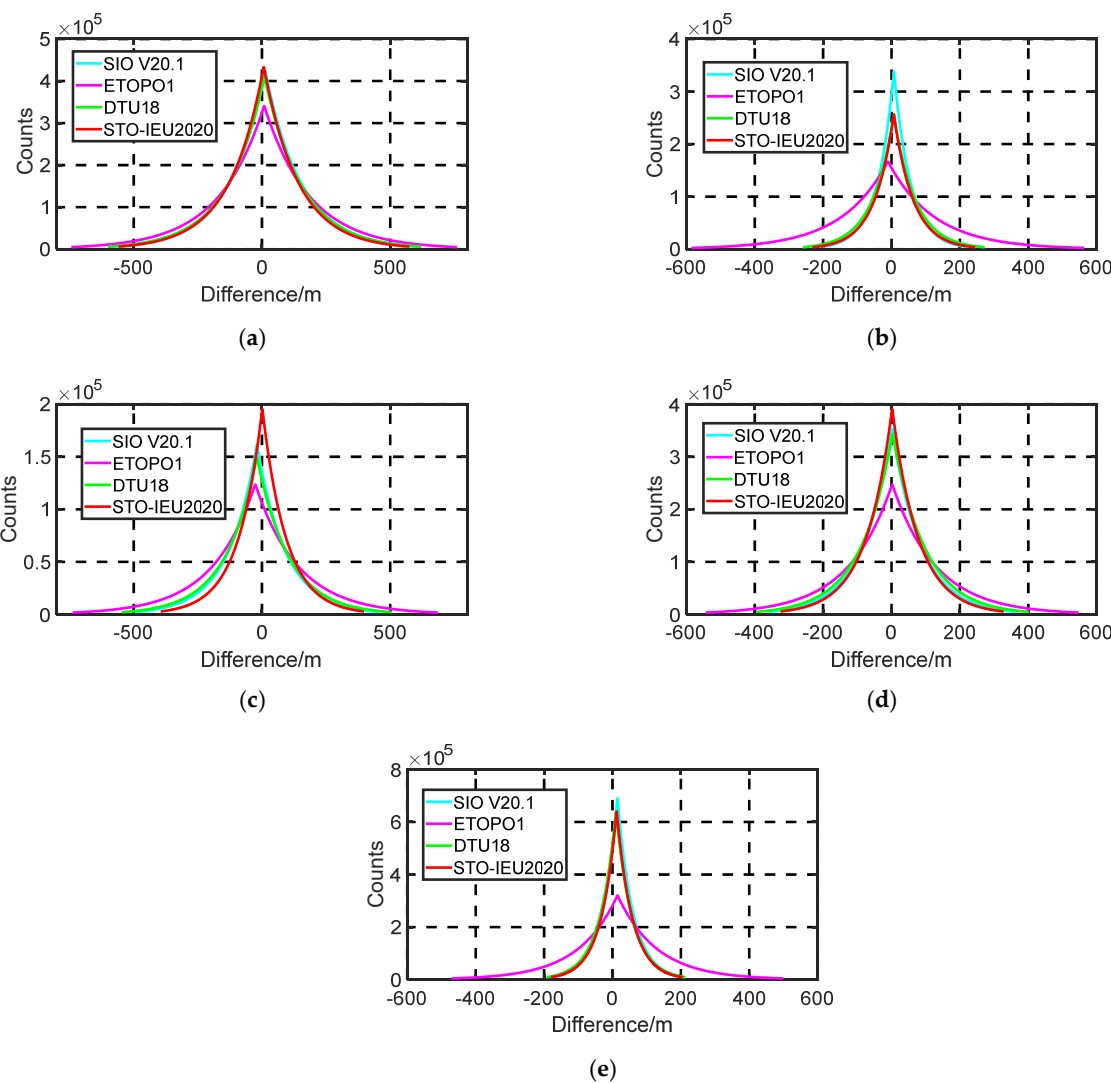

**Figure 14.** Number of checkpoints in different difference ranges (**a**) Sea Area 1; (**b**) Sea Area 2; (**c**) Sea Area 3; (**d**) Sea Area 4; and (**e**) Sea Area 5.

**Table 11.** Statistical results of difference between SIO V20.1 and STO_IEU2020 (unit: m).

| Sea Area | Max. | Min. | Mean | SD | CC | DRR | Remarks |
|---|---|---|---|---|---|---|---|
| Sea Area 1 | 3254.61 | −3029.11 | 1.10 | 112.46 | 0.9950 | 100% | PC |
| | 338.48 | −336.28 | 0.68 | 82.47 | 0.9973 | 98.29% | SC |
| Sea Area 2 | 2063.63 | −2669.50 | −5.69 | 114.71 | 0.9888 | 100% | PC |
| | 338.45 | −349.82 | −4.45 | 95.10 | 0.9921 | 98.07% | SC |
| Sea Area 3 | 2353.69 | −2271.72 | −1.30 | 157.99 | 0.9855 | 100% | PC |
| | 472.67 | −475.26 | −0.85 | 131.07 | 0.9897 | 98.04% | SC |
| Sea Area 4 | 2117.37 | −4934.82 | 5.89 | 130.40 | 0.9875 | 100% | PC |
| | 397.07 | −385.30 | 5.56 | 78.88 | 0.9950 | 97.81% | SC |
| Sea Area 5 | 2491.16 | −3483.85 | −0.62 | 92.51 | 0.9841 | 100% | PC |
| | 276.90 | −278.14 | 0.58 | 67.81 | 0.9913 | 98.02% | SC |

Note: SD denotes standard deviation; CC denotes correlation coefficient; DRR denotes data retention rate; PC denotes primary checking; and SC denotes secondary checking.

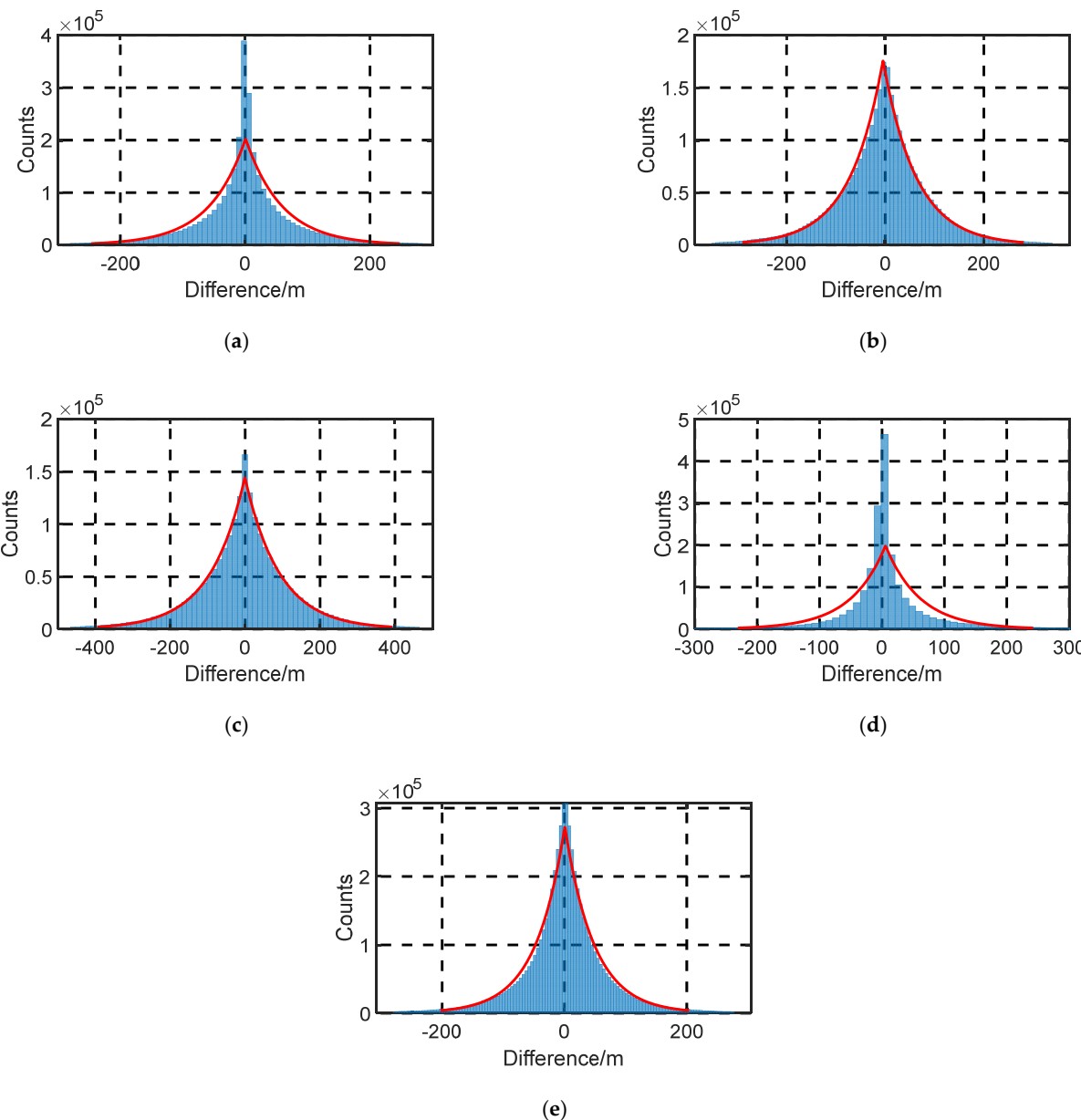

**Figure 15.** Difference between the SIO V20.1 and STO_IEU2020 at different ranges (**a**) Sea Area 1; (**b**) Sea Area 2; (**c**) Sea Area 3; (**d**) Sea Area 4; and (**e**) Sea Area 5.

## 5. Conclusions

The current limitations and deficiencies in the global ST construction include insufficient sea surface gravity input elements, weak independent performance of ST models and limited options for ST inversion. To address these challenges, this study developed a new scheme strategy for large-scale ST modeling that considers the construction efficiency of the ST model, applicability of the inversion method, heterogeneity of ST modeling environments and inversion advantages of sea surface gravity field elements. We constructed the STO_IEU2020 global ST model based on the new scheme, and assessed the results using shipborne-measured sea depth data in five evaluation areas in the Atlantic, Pacific and Indian Oceans. The evaluation results showed that within the range of ±100 m, the number of checkpoints in the STO_IEU2020 model accounted for 78.57%, 87.87%, 72.27%, 92.47% and 93.25%, with average relative accuracy better than 6%. The correlation coefficient between the STO_IEU2020 and SIO V20.1 models was above 0.99, indicating the feasibility and reliability of the large-scale ST modeling strategy proposed in this paper.

**Author Contributions:** D.F. and S.L. designed the research; D.F. and J.F. performed the research; and D.F., Y.S., Z.X. and Z.H. wrote the paper. All authors have read and agreed to the published version of the manuscript.

**Funding:** The work was supported by the Nature Science Foundation of China (42204009, 42174007, 42006197, 42074017).

**Institutional Review Board Statement:** Not applicable.

**Informed Consent Statement:** Not applicable.

**Data Availability Statement:** Not applicable.

**Acknowledgments:** The authors are grateful to the reviewers whose valuable comments significantly improved the presentation of the material.

**Conflicts of Interest:** The authors declare no conflict of interest.

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
