# Peer review of "A New Global Bathymetry Model: STO_IEU2020"

_remotesensing, doi:10.3390/rs14225744_

Round 1
Reviewer 1 Report
Please see my comments in the attached pdf file

Author Response
MAJOR COMMENTS
- Line 44, Please, explain the concept of Satellite Derived Bathymetry (SDB) that is recently quickly growing for shallow waters. Suggested reference:1109/lgrs.2020.2987778. Line 47, suggested reference: 10.1016/j.enggeo.2022.106615. Line 54, Suggested reference: 10.3390/geosciences8020063
We have added the reference such as (Albright, Glennie, 2021), (Janowski et al., 2022) and (Larry et al., 2018).
- Line 45, Vague. Explanation needed. In which areas and depths hydrographic surveys are inefficient?
We have revised the sentence according to your comments. The revised sentence is “Facing the whole ocean, ship-based hydrographic surveys have high accuracy but are inefficient and result in unevenly distributed surveys especially in the southern hemisphere.”
- “simple gross error processing was performed on the bathymetric data using S&S V20.1 bathymetry model as a priori model with a 2σ criterion.” Can you explain the processing method and why you used such a criterion?
The processing method refer to the reference (Fan, et al., 2020a). The specific process is as follows: (1) The S&S V20.1 priori bathymetry model was used to interpolate a priori bathymetry values at the ship survey position using a bilinear interpolation algorithm; (2) residual values were obtained by comparing the difference between the model interpolation results and ship survey values; and (3) absolute residual values greater than the 2-fold residual median error (approximately replaced by the Std) were removed, and quality-controlled ship survey values were obtained.
Fan D, Li S, Meng S, et al. (2020a) Applying iterative method to solving High-Order terms of seafloor topography. Marine Geodesy, 43(1): 63-85.
Usually, 3-fold residual median error (also called 3σ) is used. We used “2σ” for quality control in the manuscript would make result better.
- Figure 1. The map requires additional features - north arrow, scale bar. Figure 2, 3, as above. Figure 13. these maps are of very low quality. You will need to improve them.
We have revised according to your comments, thank you.
- “4. Construction and accuracy evaluation of global seafloor topography model” The discussion is missed? I strongly suggest to add / modify this section including a proper discussion with references to the state of the art works.
Sorry, we missed Table 10 in the first manuscript. We have added the Table 10 and proper discussion. Thank you.

Reviewer 2 Report
The research is good and encouraged.
There are some suggestions here.
1. Page 2, line 60~68, The description in this paragraph mentioned 'singular', that problem is not the target to solve in this research, it is not the main problem for the issue of inversion accuracy. It may mislead to 'nonunique' problem that is not discussed in the research.
Multivariate data could improve the reliability of the result but not the accuracy of bathymetric itself.
Other description is OK.
2. page 2, line 58, model(Hu --> missing a space.
3. page 3, line 114, (Fan et al. 2020b) --> missing 2020b.
It could be reference 10 (line 399), there is another issue, 2020b need to be put before 2020c (reference 9).
4. page 3, line 121, Fan et al. 2021a)) --> ')' doubled, removal one of them.
By the way, 2021a (reference 6) and 2021b (reference 11) can be distinguished by Fan et al. 2021 and Fan 2021. 2021a and 2021b is unnecessary.
5. page 4, line 158, 2021b --> 'b' is unnecessary.
6. page 19, line 439~441, The references 30 and 31 need be distinguished by 2020a and 2020b. The corresponding citation in the text need be changed as well.
Author Response
MAJOR COMMENTS
- Page 2, line 60~68, The description in this paragraph mentioned 'singular', that problem is not the target to solve in this research, it is not the main problem for the issue of inversion accuracy. It may mislead to 'nonunique' problem that is not discussed in the research.
We have revised the sentence according to your comments, thank you.
- page 2, line 58, model(Hu --> missing a space.
We have revised according to teacher’s suggestion. Thank you.
- page 3, line 114, (Fan et al. 2020b) --> missing 2020b.
It could be reference 10 (line 399), there is another issue, 2020b need to be put before 2020c (reference 9).
We have revised according to teacher’s suggestion. Thank you.
- page 3, line 121, Fan et al. 2021a)) --> ')' doubled, removal one of them.
By the way, 2021a (reference 6) and 2021b (reference 11) can be distinguished by Fan et al. 2021 and Fan 2021. 2021a and 2021b is unnecessary.
We have revised according to teacher’s suggestion. Thank you.
- page 4, line 158, 2021b --> 'b' is unnecessary.
We have revised according to teacher’s suggestion. Thank you.
- page 19, line 439~441, The references 30 and 31 need be distinguished by 2020a and 2020b. The corresponding citation in the text need be changed as well.
We have revised according to teacher’s suggestion. The corresponding citation in the text also changed as well. Thank you.

Reviewer 3 Report
Please see the attachment.

Author Response
Major comments:
(1)In Section 2, about the inversion methods, I suggest that the authors should briefly introduce the principles of the inversion methods.
Thank you for your comments. The inversion methods can be found in the references in the paper. In addition, our innovation does not improve the method, but introduces a new global seafloor topography model (STO_IEU2020). Therefore, we do not specifically introduce the inversion methods and formula.
(2)In Subsection 3.1.1, the data fusion of NGDC shipborne and the GEBCO_2019 multi-source bathymetry data, why the grids are divided in 15″? Please explain it.
Thank you for your comments. Because GEBCO_ 2019 is 15″ grid and STO_IEU2020 is 1' grid, to unify the input data including NGDC shipborne and GEBCO_ 2019 data (15″), the grids are divided in 15″.
(3)Where is the Table 10? It’s missing.
Thank your reminding. We have added Table 10.
(4) The input for constructing the STO_IEU2020 model is from SIO V29.1 data source, so I think it is a matter of course that the STO_IEU2020 model agrees well with the SIO V20.1. Therefore, it is recommended that the authors further discuss the impact of the data sources used on the final accuracy evaluations.
Thank you for your comments. The manuscript is mainly to introduce our design scheme to build a new global seafloor topography STO_ IEU2020. Thus, we do not research the impact of the data sources used on the final accuracy evaluations. However, “the impact of input data” is a good idea, which is worth our next special discussion in other manuscript. Thank you very much again.
Detailed comments:
- Line 92: “preset”, is “preseted” or “precise”?
Thank you. We have revised.
- Line 115: “frequency-domain inversion”, should be “Frequency-domain inversion”.
Thank you. We have revised.
- “Regression analysis is mainly suited for target sea areas with more bathymetric data and uniform distribution.” This sentence is repeated with the previous paragraph, please rephase it.
Thank you. We have revised.
- Line 148: “many”->”Many”.
Thank you. We have revised.
- Title of Figure 4: “sea” should be “Sea”.
Thank you. We have revised.
- Figure 10: It is difficult to distinguish the distributions of measured data, please redraw it. Also
the problems for Figure 13.
Thank you. We have revised.
- Line 345: When analyzing Figure 15, please first briefly introduce it.
Thank you. We have revised.
- Lines 350-351: “The findings suggest that the proposed method for global ST model construction proposed in this study is feasible…”, should be “The findings suggest that the method for global ST model construction proposed in this study is feasible…”
Thank you. We have revised.
- Reference 26: This article is in Chinese.
Thank you. We have revised.
- In addition, I suggest that the English expressions of the manuscript should be improved by a native English speaker.
Thank you. We have revised the paper again.

Reviewer 4 Report
In this article on the design and implementation of a new gravity-estimated seafloor topography model, Fan and co-authors say that they propose a new model to address current limitations in the inversion of gravity fields to produce seafloor topography models. The authors present and explain, at least in the text if not so well in their figures, a new scheme for modeling seafloor topography based on gravity. They demonstrate by comparing models in 5 areas that their model, the STO, performs about as well as the prevailing current model (SIOv20.1). Did Fan et al produce a valid, new model? Yes. Is it better or worth using compared to the previous state of the art models? No. Is this paper interesting or impactful? Unclear.
The limitations to present methods are addressed in points 1-3 on lines 60-82. In my opinion, these are not serious limitations and the outcome of the new model being similar to existing models, supports my opinion on this. Switching between GA and VGG in different areas may produce a slightly better result. However, the inversion code is likely to draw out the spatial patterns and much of the same frequency content from either dataset; hence, the lack of improvement in the final "STO" models. Model dependence on the latest SIO data isn't a major problem, and this is pretty well known. This isn't a problem and there is no solution to it. Multiple inversion methods were a bit unclear in the methods section, but it introduces a level of complication that makes implementation of the new model and confidence in it somewhat lower than the SIOv20.1 models. So, I suppose the point of this paper is simply to put forth another inversion model, and in that the authors succeed. It's unclear why this is noteworthy.
A few comments about the figures and tables. I found the red-blue scheme of the figures very difficult to see. So difficult in fact, that I'm unable to make a full evaluation of the quality of their results! I suggest revising this to standard black-on-white. The tables had inadequate captions/descriptions. I had to spend a lot of time reading and re-reading to understand what data were in the tables and from which sources and which acronymns were in use. Easily solved by putting this in the table caption.
Author Response
MAJOR COMMENTS
- They demonstrate by comparing models in 5 areas that their model, the STO, performs about as well as the prevailing current model (SIOv20.1). Did Fan et al produce a valid, new model? Yes. Is it better or worth using compared to the previous state of the art models? No. Is this paper interesting or impactful? Unclear. Multiple inversion methods were a bit unclear in the methods section, but it introduces a level of complication that makes implementation of the new model and confidence in it somewhat lower than the SIOv20.1 models. So, I suppose the point of this paper is simply to put forth another inversion model, and in that the authors succeed. It's unclear why this is noteworthy.
Thank you for your comments. We understand your comments on STO_IEU2020 global bathymetry model very well. Although STO_IEU2020 performs about as well as the prevailing current model (SIOv20.1) according to the current evaluation, the significance of our work is that we have used our own strategy to construct the global bathymetry model, and STO_IEU2020 is much better than ETOPO1 and DTU18. Just like GEBCO series models (GEBCO_08, 14, 2019, 2020, 2021) and SIO series models, they built the first version model and then updated it continuously. From this perspective, we demonstrate this ability, and we can continue to improve the model with our strategies in the future, which we think should be meaningful. We want to promote the development of global bathymetry model together.
According to your comments, we have revised the title as “A New Global Bathymetry Model: STO_IEU2020”.
- I suggest revising this to standard black-on-white. The tables had inadequate captions/descriptions. Easily solved by putting this in the table caption.
We have revised according to teacher’s suggestion. Thank you.

Round 2
Reviewer 3 Report
Thanks for the improvement made by the authors according to the review comments. I have no other suggestions.